# Hyperparameter-Free Medical Image Synthesis for Sharing Data and Improving Site-Specific Segmentation

**Alexander Chebykin**[1]                        A.CHEBYKIN@CWI.NL
**Peter A. N. Bosman**[1,2]                PETER.BOSMAN@CWI.NL
**Tanja Alderliesten**[3]                 T.ALDERLIESTEN@LUMC.NL

[1] *Centrum Wiskunde & Informatica, Amsterdam, The Netherlands*

[2] *Delft University of Technology, Delft, The Netherlands*

[3] *Leiden University Medical Center, Leiden, The Netherlands*

**Editors:** Accepted for publication at MIDL 2024

## Abstract

Sharing synthetic medical images is a promising alternative to sharing real images that can improve patient privacy and data security. To get good results, existing methods for medical image synthesis must be manually adjusted when they are applied to unseen data. To remove this manual burden, we introduce a **Hy**perparameter-**Free** distributed learning method for automatic medical image **S**ynthesis, **S**haring, and **S**egmentation called **HyFree-S3**. For three diverse segmentation settings (pelvic MRIs, lung X-rays, polyp photos), the use of HyFree-S3 results in improved performance over training only with site-specific data (in the majority of cases). The hyperparameter-free nature of the method should make data synthesis and sharing easier, potentially leading to an increase in the quantity of available data and consequently the quality of the models trained that may ultimately be applied in the clinic. Our code is available at https://github.com/AwesomeLemon/HyFree-S3.

**Keywords:** Synthetic data, Distributed Learning, AutoML, Segmentation.

## 1. Introduction

Deep learning methods can be beneficial for medical applications, but often suffer from limited data availability (Bowles et al., 2018). Generating and sharing synthetic datasets was suggested as a viable solution (Dube and Gallagher, 2014).

Many approaches can synthesize medical images (Bowles et al., 2018; Yi et al., 2019), fewer jointly produce segmentation maps (Guibas et al., 2018; Han et al., 2023) (which would be required for training down-stream segmentation models) and, to the best of our knowledge, not a single method exists that could be applied to a new task without manually adjusting hyperparameters of training or data preprocessing.

The benefits of a hyperparameter-free adaptive method are self-evident. Such a method was realized for segmentation tasks by nnU-Net (Isensee et al., 2021), a method of automatically adjusting architecture and hyperparameters of a U-Net (Ronneberger et al., 2015) that showed excellent and robust performance in tens of challenges (Isensee et al., 2021).

Automatic adaptation of a high-quality underlying model and training pipeline is a general idea that could be extended to other tasks, such as medical image synthesis. A Generative Adversarial Network (GAN) called StyleGAN2 (Karras et al., 2020) demonstrated

good results in medical image synthesis (Woodland et al., 2022). However, it currently does not automatically adapt to the image dimensions or the dataset size.

In this paper, we introduce an automatically adjustable StyleGAN2 setup and integrate it with nnU-Net to create a **Hy**perparameter-**Free** medical image **S**ynthesis, **S**haring, and **S**egmentation method called **HyFree-S3**. We construct it as a distributed learning method where each site (e.g., a hospital) can automatically and asynchronously create a synthetic dataset and share it. A segmentation model can be automatically trained on the merged synthetic data and distributed back to the sites to be further automatically fine-tuned for improved performance on local data (see Figure 1).

This approach to distributed learning has practical advantages of requiring minimal co-ordination between sites and of reduced privacy risk thanks to not sharing real data, models trained with it, or their gradients (which is a potential source of data leakage in federated learning (Zhu et al., 2019)). An important concern is whether synthetic data includes memorized real data, which we address with a quantitative and qualitative investigation, as well as a technique for ensuring that synthetic data is not too similar to the real data.

We evaluate our method in three segmentation settings (pelvic MRIs, lung X-rays, polyp photos) to test its generality, the impact of synthetic data sharing, and the difference in performance compared to the realistic baseline of using only local data and the strong baseline of having central access to all the real data.

In this paper, we only consider 2D models: compared to 3D models, they have lower computational requirements and need less data to be trained (which is important for the data sharing setting where some sites could have small datasets). In the future, our approach could be extended to 3D models for improved segmentation performance in settings where there is enough data and computational resources.

The contributions of this paper are as follows:
- We propose HyFree-S3, a hyperparameter-free distributed learning method integrating image synthesis, data sharing, and segmentation.
- Towards that goal, we introduce a hyperparameter-free StyleGAN2 setup that can adapt to various image dimensions and dataset sizes.
- The segmentation quality of HyFree-S3 is evaluated in three settings. Furthermore, its scaling behavior and ability to avoid data memorization is investigated.

## 2. Related work

### 2.1. Sharing synthetic medical image data

Synthesizing medical data for sharing purposes so as to avoid privacy issues is an established idea in the literature (Goncalves et al., 2020). Medical *image* data can be synthesized by GANs (Bowles et al., 2018; Yi et al., 2019; Sun et al., 2022), but for supervised learning to be possible, an annotation needs to be associated with a synthetic image. While conditioning GANs on an image class is straightforward when generating data for classification tasks (Hu et al., 2018), conditioning on segmentation maps (Chang et al., 2023) to generate data for segmentation tasks requires the additional complexity of sharing segmentations to condition on (e.g., by training a separate GAN for them (Guibas et al., 2018; Han et al., 2023)). Thambawita et al. (2022) propose a GAN for joint unconditional generation of polyp images and segmentations that improves performance but requires a separate GAN to be trained for each input image, likely impeding scaling to larger datasets.

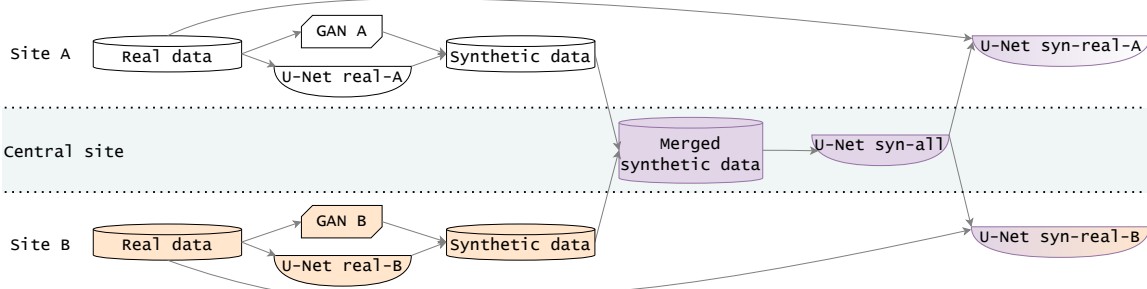

Figure 1: Overview of HyFree-S3 for two sites. Synthetic datasets are generated at each site independently, merged at a central site, and used in training a general segmentation model. That model is copied to all sites and independently fine-tuned on the local data. All models automatically adapt to the properties of the data.

## 2.2. Distributed learning with medical image data

Federated learning (Konečný et al., 2016) can be applied to medical imaging data (Rieke et al., 2020; Adnan et al., 2022) where it allows the global model to be trained on diverse data from multiple hospitals (Ng et al., 2021) without sharing local data.

Federated learning algorithms for training a GAN (Rasouli et al., 2020) can be applied to medical data (Chang et al., 2023) but they incur privacy costs because real data could be reconstructed from the gradients passed between sites (Zhu et al., 2019). A common solution (Chang et al., 2020, 2023; Wang et al., 2023) is to train only the generator globally while the discriminators are trained per-site using the local data and the synthetic data from the global generator.

Our method differs from the existing distributed learning techniques in three ways. Firstly, it is asynchronous and does not require simultaneous online access to sites. Secondly, the generated data is filtered to not contain memorized data (in contrast to sharing a black-box generative neural network with potentially undiscovered vulnerabilities). Finally, HyFree-S3 is adaptable and hyperparameter-free, thus making it potentially easier to use.

## 3. Method

### 3.1. Overview of the method

We assume that $N$ sites (such as medical centers) have a goal of solving a segmentation task. Each site has a local dataset that cannot be straightforwardly shared due to privacy or security concerns. The datasets may differ in sizes and image characteristics per-site.

Figure 1 shows the flow of data and models in HyFree-S3. Firstly, each site runs hyperparameter-free methods to create a generative model and a segmentation model. A generative model (Section 3.3) is used to create data without segmentations, which are then segmented by the segmentation model (Section 3.2) to create a complete synthetic dataset for sharing. The reasoning behind using two separate models is given in Section 3.4.

Next, the synthetic datasets from all sites are merged in a central location, and a general segmentation model is trained using all the synthetic data. This general segmentation model is transferred back to the sites, and is further automatically fine-tuned at each one. The resulting models benefit from the general pretraining but are specialized for each site.

### 3.2. Hyperparameter-free medical image segmentation

nnU-Net is a robust medical image segmentation method that was shown to perform excellently in a wide variety of competitions and benchmarks (Isensee et al., 2021). nnU-Net can adapt to diverse datasets without hyperparameter tuning thanks to heuristics for adjusting the underlying U-Net architecture (Ronneberger et al., 2015) and the training procedure. The spatial dimensions of the data influence the input size of the model, depths of the encoder (decoder), downsampling (upsampling) strides, and convolution kernel sizes.

To adjust the hyperparameters to the fine-tuning setting not considered by Isensee et al. (2021), we add linear learning rate warm-up for the first 10% of training epochs (Mosbach et al., 2020) and do not otherwise change the default hyperparameters.

### 3.3. Hyperparameter-free data synthesis

StyleGAN2 (Karras et al., 2020) is a powerful generative model that by default generates square images of a resolution of a power of two. However, medical images come in a variety of resolutions. Ideally, synthetic images of the appropriate resolution should be generated.

For segmentation tasks, nnU-Net adapts the structure of the U-Net to the resolution. We noticed a similarity between the structures of the generator/discriminator of a GAN and those of the decoder/encoder of a U-Net. Both the GAN generator and the U-Net decoder gradually upscale a low-resolution many-channel latent representation of an image towards a high-resolution few-channel output. The architectures need to strike the correct balance between the speeds of increasing the resolution and decreasing the number of channels. As such, a network that strikes the correct balance in one setting, seems likely to do so in the other. Analogously, the discriminator and the encoder gradually downscale a high-resolution few-channel input towards a low-resolution many-channel representation.

Therefore, it appears natural to reuse the hyperparameters of the encoder and the decoder automatically determined by nnU-Net (depths of the networks, convolution strides, and kernel sizes) for the discriminator and the generator of a GAN. This will allow it to create non-square non-power-of-two-sized images of the exact size determined by nnU-Net.

Next, the number of training steps, $n_{steps}$, needs to be set automatically. In StyleGAN2, $n_{steps}$ is defined in thousands of real images processed during training. As the dataset size increases, so should $n_{steps}$ (to allow the GAN to learn from a larger amount of data). Setting $n_{steps}$ to the number of images in the dataset ensures good image quality across different dataset scales, keeps training times short, and prevents memorization (see Section 4.3.3).

The number of images to be generated, $n_{gen}$, also needs to be determined. While generating images with GANs requires little compute and time, generating increasingly large numbers of samples leads to diminishing returns (Ravuri and Vinyals, 2019). We also need to consider the proportions of synthetic data coming from different sites used in training the general segmentation model: generative models trained on larger datasets should contribute more than those trained on smaller datasets, as they likely have higher quality. For these reasons, $n_{gen}$ is set to ten times the dataset size for each dataset.

We use the augmentations setup of Zhao et al. (2020) that was shown to improve performance and help avoid overfitting with datasets as small as 100 images.

### 3.4. Why not synthesize images and segmentations jointly?

Segmenting with a separate model was a deliberate design choice and constraint. While it is possible to train a generative model to output segmentations as well as images, this would lead to segmentations influencing the images. This is undesirable for medical data sharing, as the references in many segmentation scenarios vary due to the protocol and observer variation. Letting these variations influence the generated *images* should be avoided: then the images themselves can still contribute to the improvement of the models (e.g., during unsupervised pretraining) even if segmentations need to be discarded or redone. Additionally, if no annotations are available, HyFree-S3 can still be utilized to share images (unlike methods that condition on segmentations).

### 3.5. Measuring memorization and preventing real data leakage

Synthetic data should be similar enough to the real data for the models trained on one to transfer to the other, and dissimilar enough for the privacy concerns to be alleviated. Generative models are capable of memorizing their training data and outputting it as "synthetic" samples (Feng et al., 2021). Memorization is difficult to determine because the reproduction typically includes some variation or noise. It is not obvious where the threshold between the presence and the absence of memorization should be.

We propose automatically determining this threshold based on the real data itself as follows: firstly, the patients are randomly split into two subsets. For each image in the first subset, its dissimilarity with each image in the second subset is computed using the L2 distance between their OpenCLIP embeddings (Ilharco et al., 2021). The minimal dissimilarity (i.e., the distance to the nearest neighbour) is stored for each image. The threshold is then defined as the $p$-th percentile of these dissimilarities.

After the synthetic dataset is generated, the dissimilarity of each synthetic image to all real images is similarly computed. If the distance of a synthetic image to its nearest real-image neighbor is below the determined threshold, the image is declared to be memorized and is discarded. For $p = 0$ (the minimum of dissimilarities), this procedure ensures that any synthetic image is only as similar to a real image as one unrelated real image to another; we set $p = 5$ to guard against outliers. The procedure relies on the quality of the embedding model that can only be demonstrated empirically (Cherti et al., 2023).

## 4. Experiments

### 4.1. Experiment setup

In our experiments, we emulate a distributed setting with $N$ sites. As per Section 3.1, for each site $S_i$, GAN-$S_i$ and U-Net-*real*-$S_i$ are trained and used to generate a synthetic dataset. U-Net-*syn-all* is trained on the merged datasets and fine-tuned at each site to get U-Net-*syn-real*-$S_i$. U-Net-*real-all* is trained on merged real data as a baseline. All the experiments are performed via five-fold cross-validation (3 folds for training, 1 fold for validation and test each). The mean and the standard deviation of the Dice Score (DS) and the 95th percentile of the Hausdorff Distance (HD95) across the folds are reported. All the evaluations were performed on real data. The results of statistical testing are given in Appendix A. Appendix D contains comparisons with federated learning baselines. The ablations of pretraining with single-site synthetic data and of using the standard StyleGAN2 architecture are reported in Appendices F, G.

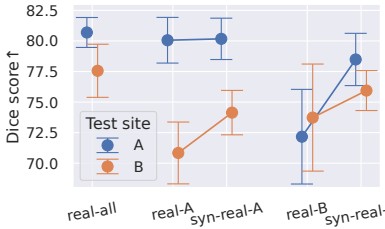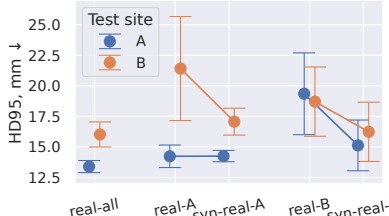

Figure 2: Results for the **Cervix** data: U-Nets trained with the settings specified in Section 4.1 and evaluated on sites A and B (5 folds).

## 4.2. Datasets

**Cervix**: a private dataset from the Leiden Univercity Medical Center consisting of T2-weighted MRI scans of 185 cervical cancer patients who underwent brachytherapy, with 4 organs-at-risk (bladder, bowel, rectum, sigmoid) delineated. The dataset was split into two sites based on the scanner to emulate a non-i.i.d. data distribution: site A (Philips Ingenia 1.5T (128 patients)), and site B (Philips Intera (36 patients), Ingenia 3T (13), Achieva (8)). The median resolution is $37 \times 432 \times 432$ voxels, the median spacing is $4 \times 0.53 \times 0.53$ mm.

**Lung**: QaTa-COV19 (Degerli et al., 2022), a dataset of COVID-19 chest X-ray images and binary segmentation masks of pneumonia. We use 6,307 images for which an anonymized patient ID is provided (2,130 patients) and randomly split patients into 2 or 8 sites. The median resolution is $224 \times 224$ pixels.

**Polyp**: polyp photos with binary segmentation masks of polyps, site A contains data from HyperKvasir (Borgli et al., 2020) (1000 images, median resolution $530 \times 621$ pixels), site B contains data from CVC-ClinicDB (Bernal et al., 2015) (612 images, $384 \times 288$ pixels).

## 4.3. Results

### 4.3.1. SYNTHETIC DATA SHARING LEADS TO IMPROVED SEGMENTATION QUALITY

Figure 2 shows that in the experiments with **Cervix**, DS and HD95 metrics improve in most cases. The performance on site A is approximately the same across the *real-A*, *syn-real-A*, *real-all* settings, showing that adding data from site B is not very helpful even if it is real data. This is likely due to the large number and uniformity of patients in A itself. Nonetheless, the performance in the *syn-real-A* setting on site B is improved compared to *real-A* (by 3.3 DS and 4.3 HD95 on average), showing that pretraining on merged synthetic data improves the robustness of the model to data shifts. For site B, the pretrained and fine-tuned model (*syn-real-B*) outperforms its counterpart trained exclusively on the local data (*real-B*) when tested on both A (by 6.3 DS and 4.3 HD95 on average) and B (by 2.2 DS and 2.5 HD95 on average). The full results for all datasets are given in Appendix B.

It can be seen in Figure 3 that switching from the model trained only on local data (*real*) to the one pretrained on the merged synthetic data and fine-tuned on local data (*syn-real*) leads to improvements in the **Lung** setting of 0.8 DS on average. For **Polyp**, the improvement for the target site is minor (0.5 DS on average), but the improvement in robustness to data shifts, as measured by the performance on the other site, is large (on average, 2.7 DS for A and 13.4 DS for B).

While training on real data centrally (*real-all*) gives the best results overall, our method comes close (the largest difference is 2.0 DS in **Polyp**-B) without requiring real data sharing.

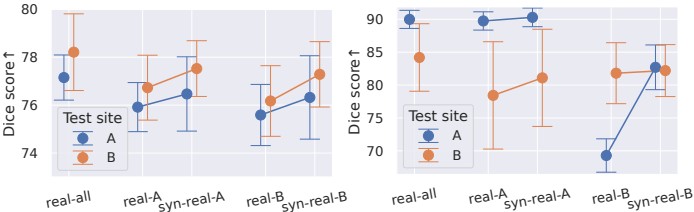
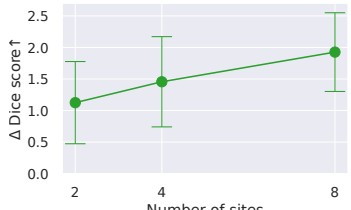

Figure 3: DS for the **Lung** (left) and **Polyp** (right) data: U-Nets trained in the settings specified in Section 4.1 and evaluated on sites A and B (5 folds).

Figure 4: DS improvement of *syn-real* over *real* as more sites are added (**Lung**).

### 4.3.2. BENEFITS OF SYNTHETIC DATA SHARING INCREASE WITH MORE SITES

The scaling behavior of HyFree-S3 is investigated in the **Lung** setting, with the data split into 8 sites. The general segmentation model is pretrained with the synthetic data from 2, 4, or 8 sites, and then fine-tuned at each site. The average difference in DS between *real* (training with local data only) and *syn-real*, shown in Figure 4, becomes larger as the number of site grows, showing that the method is scalable and enables larger improvements when more sites join.

### 4.3.3. MEMORIZATION IS NOT OBSERVED

Per Section 3.5, we use OpenCLIP embeddings to compare either subsets of real images, or real images and synthetic images. As a sanity check, we established that an image present in two sets will be its own nearest neighbor, and that a mirrored image will most often be the nearest neighbor of the original image (in 97.9% of cases for **Lung**).

Figure 5 (top right) shows a histogram of distances to the nearest neighbor when comparing two subsets of real images (from a **Lung** experiment), as well as their $5^{\text{th}}$ percentile that will be used as a threshold to filter synthetic images. The histogram of distances between real and synthetic images in Figure 5 (bottom right) has a similar shape but shifted to the right. As a result, all synthetic images have a distance above the threshold, meaning that while synthetic images are generally similar to real images, they are not too similar.

We visualize adversarially chosen real and synthetic images in Figure 5 (left). We select two synthetic images (column 2) with the smallest nearest neighbor distances to some real images (column 1), and the synthetic images that have the second-smallest distances to these real images (column 3). The images are broadly similar but clearly distinct and do not demonstrate memorization (it should be noted that our memorization analysis relies on OpenCLIP embeddings, see Appendix E for further discussion of memorization).

For **Lung** and **Polyp**, only 6 synthetic images (out of $3 \times 10^5$) are discarded as too similar to real images. For **Cervix**, 1,180 images (out of $2.6 \times 10^5$) are discarded, some of which look very similar to their real nearest neighbours but none are exact duplicates, which is consistent with the nature of 3D data where nearby slices are similar. See Appendix C for visual comparisons between real and synthetic images. Only 0.2% of synthetic images being discarded shows that our GANs typically do not memorize images but the proposed filtering scheme could nonetheless help to avoid sharing potentially memorized data.

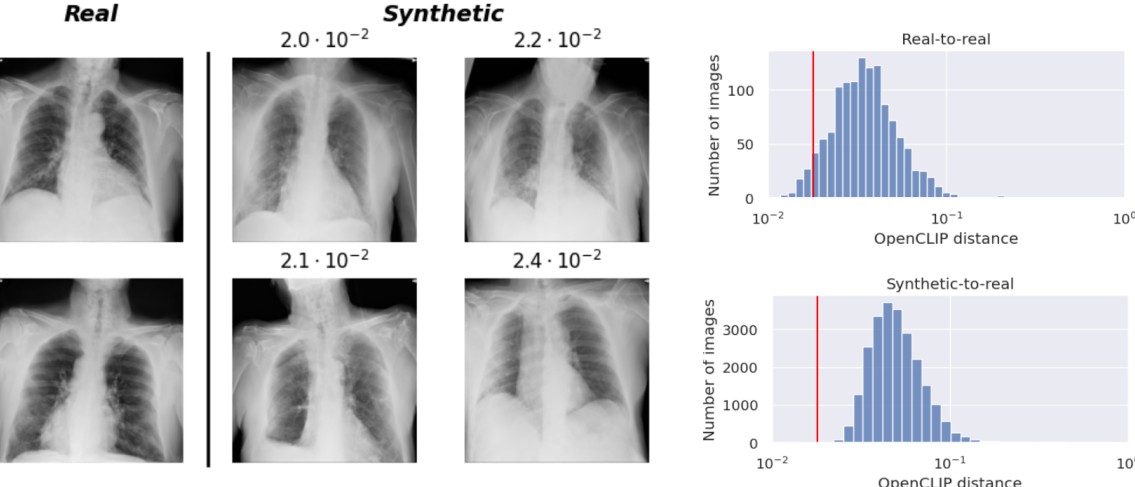

Figure 5: **Left:** real images that are the closest to any synthetic image and the two closest synthetic images (including the distance to the real image). **Right:** a distribution of distances to the nearest neighbor for *(top)* two subsets of real images or *(bottom)* synthetic and real images, and the $5^{th}$ percentile of the real-to-real distances.

## 5. Discussion & Conclusion

We have introduced HyFree-S3, a method for synthesizing medical images for sharing and segmentation that does not require adjusting hyperparameters, as it relies on our hyperparameter-free setup of StyleGAN2 for image generation and similarly hyperparameter-free nnU-Net (Isensee et al., 2021) for segmentation.

The method is asynchronous and does not expect different sites to share infrastructure or to be online simultaneously. HyFree-S3 was shown to come close to the performance of training with centralized real data and to improve upon training only with local data, while not requiring sharing real patient data. The synthetic dataset produced by HyFree-S3 could be reused for training future models without going back to the sites where the real data is stored. We additionally proposed a memorization evaluation technique for ensuring that synthetic data is sufficiently dissimilar from real data.

However, our method also has a number of limitations. Firstly, as models for data sharing are trained on each site independently, large enough datasets have to be present there (especially for training a GAN). Secondly, we generate images and annotations separately for reasons described in Section 3.4, which nonetheless means that the annotations produced by an independent model will not perfectly correspond to the images, introducing noise to synthetic data. Finally, our method as presented cannot be applied to all possible datasets, as it relies on nnU-Net for preprocessing and segmentation: nnU-Net can handle a variety of inputs but not, e.g., high-resolution histopathology images. However, extending nnU-Net with an appropriate data preprocessing procedure would make HyFree-S3 applicable too.

Despite some limitations, adaptable methods that do not require task-specific tuning are still valuable for the translation of deep learning models into the real world, where there may not be time or expertise for careful manual adaptation to each task.

## Acknowledgments

This work is part of the research project DAEDALUS which is funded via the Open Technology Programme of the Dutch Research Council (NWO), project number 18373; part of the funding is provided by Elekta and ORTEC LogiqCare. We acknowledge the use of ChatGPT for grammar assistance in the preparation of this manuscript.

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

## Appendix A. Statistical testing

Table 1 lists p-values for one-sided Wilcoxon pairwise rank tests (Wilcoxon, 1992) with Bonferroni correction (Dunn, 1961). For each dataset, we compare metrics of *syn-real* (pretraining with merged synthetic data followed by site-specific fine-tuning on real data) against that of *real* (training with local real data only) with the null hypothesis that *syn-real* performs worse. The performances of *syn-real-A* (evaluated on A, B) and *syn-real-B* (evaluated on A, B) across 5 folds are pulled together for increased sample size (and similarly for *real-A* and *real-B*). The total sample size is 20 for each test. P-values below the significance threshold of 0.0025 are highlighted (target p-value= 0.01, 4 tests, corrected $p = 0.0025$). In all cases the null hypothesis is rejected, thus we can conclude that *syn-real* performs statistically significantly better than *real*.

Table 1: P-values of conducted experiments (rounded to four decimal places).

| Dataset | Metric | P-value |
|---------|--------|---------|
| **Cervix** | DS | 0.0006 |
| **Cervix** | HD95 | 0.0007 |
| **Lung** | DS | 0.0002 |
| **Polyp** | DS | 0.0002 |

## Appendix B. Full results

Tables 2, 3, 4 give full results for our **Cervix**, **Lung**, **Polyp** experiments. For **Cervix** and **Lung**, U-Net-syn-$S_i$ are additionally trained on local synthetic datasets to compare with U-Net-syn-all. As expected, they perform worse.

Table 5 gives per-organ performance for the **Cervix** dataset. Figure 6 contains slice predictions for qualitative comparison of a *real* and a *syn-real* models.

Table 2: Results for the **Cervix** data. Each column corresponds to a U-Net trained in the specified setting (see Section 4.1) and evaluated on sites A and B. Mean $\pm$ st. dev. are reported for 5 folds.

| Metric | Test site | Training setting | | | | | | | |
|--------|-----------|---------|---------|---------|---------|-----------|---------|---------|------------|
| | | real-all | syn-all | real A | syn A | syn-real A | real B | syn B | syn-real B |
| DS | A | $80.7 \pm 1.2$ | $79.4 \pm 1.3$ | $80.1 \pm 1.9$ | $79.0 \pm 1.5$ | $80.2 \pm 1.7$ | $72.2 \pm 3.9$ | $72.4 \pm 3.8$ | $78.5 \pm 2.1$ |
| | B | $77.6 \pm 2.2$ | $75.4 \pm 2.6$ | $70.8 \pm 2.5$ | $69.9 \pm 1.5$ | $74.1 \pm 1.8$ | $73.7 \pm 4.4$ | $72.8 \pm 3.9$ | $75.9 \pm 1.6$ |
| HD95 | A | $13.4 \pm 0.5$ | $14.8 \pm 1.1$ | $14.2 \pm 0.9$ | $15.3 \pm 1.2$ | $14.3 \pm 0.5$ | $19.4 \pm 3.3$ | $19.9 \pm 3.1$ | $15.1 \pm 2.1$ |
| | B | $16.0 \pm 1.0$ | $16.1 \pm 2.7$ | $21.4 \pm 4.3$ | $21.0 \pm 3.9$ | $17.1 \pm 1.1$ | $18.7 \pm 2.8$ | $19.0 \pm 3.0$ | $16.2 \pm 2.4$ |

Table 3: Results for the **Lung** data. Each column corresponds to a U-Net trained in the specified setting (see Section 4.1) and evaluated on sites A and B. Mean ± st. dev. are reported for 5 folds.

| Metric | Test site | Training setting | | | | | | | |
|---|---|---|---|---|---|---|---|---|---|
| | | real-all | syn-all | real A | syn A | syn-real A | real B | syn B | syn-real B |
| DS | A | $77.2 \pm 0.9$ | $75.8 \pm 1.1$ | $75.9 \pm 1.0$ | $74.7 \pm 1.3$ | $76.5 \pm 1.6$ | $75.6 \pm 1.3$ | $74.6 \pm 1.5$ | $76.3 \pm 1.7$ |
| | B | $78.2 \pm 1.6$ | $76.7 \pm 1.3$ | $76.7 \pm 1.4$ | $75.3 \pm 1.7$ | $77.5 \pm 1.2$ | $76.2 \pm 1.5$ | $75.0 \pm 1.6$ | $77.3 \pm 1.4$ |

Table 4: Results for the **Polyp** data. Each column corresponds to a U-Net trained in the specified setting (see Section 4.1) and evaluated on sites A and B. Mean ± st. dev. are reported for 5 folds.

| Metric | Test site | Training setting | | | | | |
|---|---|---|---|---|---|---|---|
| | | real-all | syn-all | real A | syn-real A | real B | syn-real B |
| DS | A | $90.0 \pm 1.4$ | $87.7 \pm 1.4$ | $89.7 \pm 1.4$ | $90.3 \pm 1.4$ | $69.3 \pm 2.5$ | $82.7 \pm 3.4$ |
| | B | $84.2 \pm 5.1$ | $80.9 \pm 5.7$ | $78.4 \pm 8.2$ | $81.1 \pm 7.4$ | $81.8 \pm 4.6$ | $82.2 \pm 3.9$ |

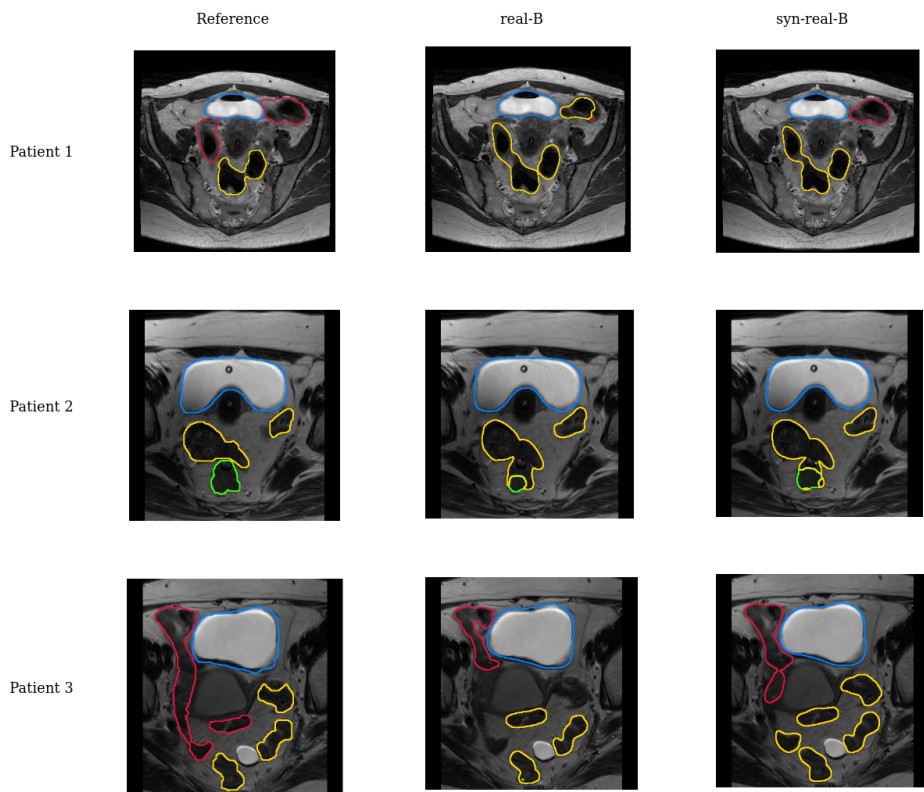

Figure 6: Comparison of *real-B* to *syn-real-B* using slices from 3 random patients. Depicted are bladder (blue), bowel (red), rectum (green), sigmoid (yellow).

| Model | Avg. | | Bladder | | Bowel | | Rectum | | Sigmoid | |
|---|---|---|---|---|---|---|---|---|---|---|
| | A | B | A | B | A | B | A | B | A | B |
| Dice Score | | | | | | | | | | |
| real-A | 80.1 ±1.9 | 70.8 ±2.5 | 96.0 ±0.4 | 91.9 ±2.5 | 68.8 ±3.5 | 56.0 ±6.4 | 81.7 ±2.9 | 76.8 ±3.3 | 73.7 ±2.5 | 58.7 ±3.4 |
| syn-A | 79.0 ±1.5 | 69.9 ±1.5 | 96.0 ±0.4 | 91.7 ±2.5 | 66.6 ±2.8 | 53.2 ±4.9 | 81.7 ±2.8 | 76.5 ±2.8 | 71.7 ±2.0 | 58.2 ±2.0 |
| syn-real-A | 80.2 ±1.7 | 74.1 ±1.8 | 96.0 ±0.4 | 93.3 ±1.3 | 69.6 ±2.8 | 61.4 ±6.3 | 81.5 ±2.8 | 79.3 ±3.6 | 73.5 ±2.0 | 62.6 ±3.1 |
| real-B | 72.2 ±3.9 | 73.7 ±4.4 | 95.3 ±0.6 | 94.1 ±0.9 | 53.2 ±8.5 | 60.3 ±9.9 | 77.4 ±3.1 | 79.1 ±3.2 | 62.8 ±6.4 | 61.5 ±7.0 |
| syn-B | 72.4 ±3.8 | 72.8 ±3.9 | 95.4 ±0.5 | 93.9 ±1.2 | 53.4 ±8.9 | 57.7 ±11.0 | 77.5 ±2.8 | 78.8 ±3.5 | 63.3 ±6.5 | 60.8 ±3.8 |
| syn-real-B | 78.5 ±2.1 | 75.9 ±1.6 | 95.8 ±0.4 | 93.9 ±1.2 | 65.6 ±3.9 | 63.4 ±5.6 | 81.5 ±2.7 | 80.5 ±3.3 | 71.0 ±3.7 | 66.0 ±3.7 |
| real-all | 80.7 ±1.2 | 77.6 ±2.2 | 96.2 ±0.4 | 94.4 ±1.0 | 70.1 ±2.1 | 65.5 ±6.5 | 81.8 ±2.6 | 82.4 ±2.5 | 74.7 ±1.6 | 68.0 ±3.0 |
| syn-all | 79.4 ±1.3 | 75.4 ±2.6 | 96.1 ±0.3 | 94.0 ±1.2 | 67.5 ±2.2 | 61.9 ±7.9 | 82.0 ±2.8 | 81.0 ±3.2 | 72.1 ±2.0 | 64.9 ±3.8 |
| Hausdorff Distance (95th percentile) | | | | | | | | | | |
| real-A | 14.2 ±0.9 | 21.4 ±4.3 | 3.6 ±1.0 | 11.7 ±8.4 | 20.3 ±1.8 | 25.5 ±2.7 | 13.5 ±2.5 | 16.4 ±4.9 | 19.6 ±3.3 | 32.1 ±7.8 |
| syn-A | 15.3 ±1.2 | 21.0 ±3.9 | 3.8 ±1.0 | 13.9 ±9.9 | 22.0 ±3.4 | 26.6 ±4.8 | 13.6 ±2.1 | 14.1 ±3.2 | 21.9 ±3.8 | 29.2 ±6.7 |
| syn-real-A | 14.3 ±0.5 | 17.1 ±1.1 | 3.7 ±1.3 | 7.9 ±4.3 | 18.7 ±3.6 | 20.6 ±2.4 | 13.8 ±2.3 | 12.7 ±2.3 | 20.9 ±2.2 | 27.1 ±5.0 |
| real-B | 19.4 ±3.3 | 18.7 ±2.8 | 3.9 ±0.5 | 5.8 ±2.2 | 28.7 ±6.6 | 24.6 ±6.1 | 17.0 ±3.3 | 15.2 ±2.5 | 27.8 ±5.6 | 29.3 ±4.7 |
| syn-B | 19.9 ±3.1 | 19.0 ±3.0 | 4.4 ±1.1 | 5.9 ±2.5 | 28.5 ±7.6 | 26.7 ±8.1 | 17.4 ±3.8 | 14.4 ±2.5 | 29.2 ±5.1 | 29.2 ±4.2 |
| syn-real-B | 15.1 ±2.1 | 16.2 ±2.4 | 4.1 ±0.8 | 6.1 ±2.8 | 22.8 ±5.8 | 21.2 ±3.3 | 13.5 ±1.7 | 13.3 ±2.4 | 20.2 ±5.0 | 24.4 ±6.8 |
| real-all | 13.4 ±0.5 | 16.0 ±1.0 | 2.9 ±0.4 | 5.4 ±2.1 | 18.8 ±3.5 | 20.9 ±3.2 | 13.4 ±2.0 | 12.3 ±2.6 | 18.5 ±3.0 | 25.5 ±3.1 |
| syn-all | 14.8 ±1.1 | 16.1 ±2.7 | 3.9 ±1.1 | 5.5 ±2.4 | 20.9 ±3.7 | 20.8 ±3.4 | 13.7 ±2.0 | 12.4 ±2.2 | 20.6 ±4.3 | 25.5 ±7.6 |

Table 5: Per-organ metrics for the **Cervix** data. Each row corresponds to a U-Net trained in the specified setting (see Section 4.1) and evaluated on sites A and B. Mean ± st. dev. are reported for 5 folds.

## Appendix C. Visually checking memorization of synthetic images

In Figure 7, we provide examples of synthetic images that were too similar to real images and therefore discarded. Visually, the synthetic images do not appear to be exact copies of real images.

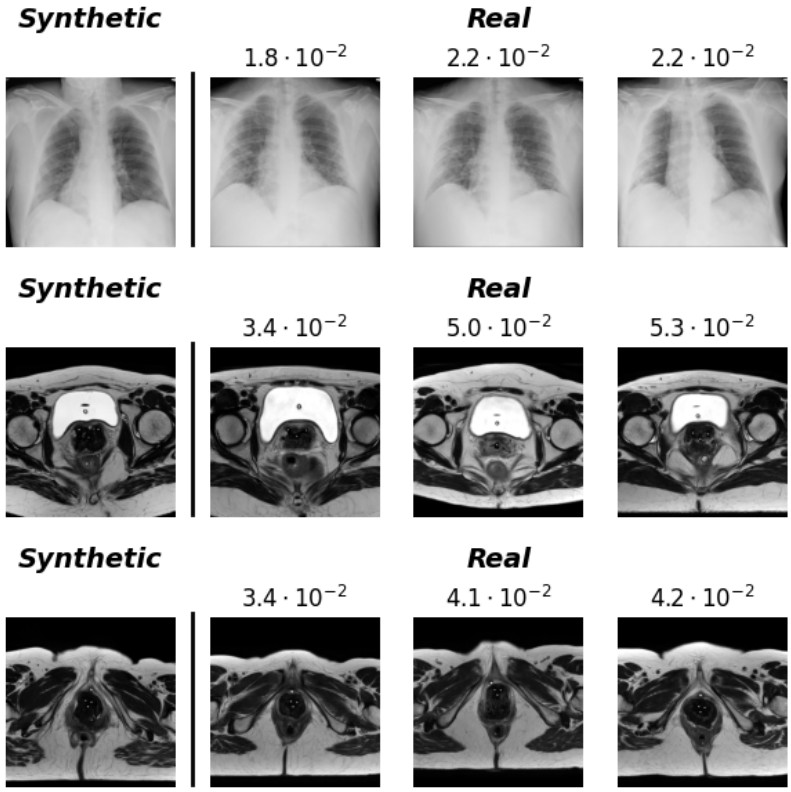

Figure 7: Examples of discarded synthetic images and three nearest-neighbor real images for each (annotated with distances to the synthetic image).

## Appendix D. Federated learning baselines

To compare our approach to federated learning, we run two federated learning baselines: Federated Averaging (FedAvg) (McMahan et al., 2017) and Distributed Synthetic Learning (DSL) (Chang et al., 2023).

FedAvg trains a segmentation model on the real data at each location and periodically averages the weights to get a global model. We implement FedAvg atop nnU-Net for fair comparison to our approach. DSL is a state-of-the-art approach to training a GAN in a federated manner to be used for generating synthetic data and training a U-Net. Our DSL experiments are based on the official implementation of DSL, see our fork at https://github.com/AwesomeLemon/DSL_All_Code.

Since the key contribution of our method is its automatic and hyperparameter-free nature, we run the baselines in a similar setting of no hyperparameter tuning. For FedAvg, we used 1000 communication rounds with 1 epoch of local training in between since it was the best setting in (McMahan et al., 2017). For DSL, we followed the paper (Chang et al., 2023) and the official code base. For hyperparameters that differed across the three datasets in DSL, we used the median values: $\lambda_{L1}$: 300, 150, 100 → 150; batch size: 6, 6, 3 → 6).

The results of the experiments are reported in Tables 6, 7, 8. FedAvg on average has 5.3 worse Dice Score (DS) than HyFree-S3 in the i.i.d. setting of **Lung**, with the gap widening further in the non-i.i.d settings: 10.7 DS for **Cervix**, 29.2 DS for **Polyp**. DSL performs substantially worse in all settings (on average: **Lung**: 32.8 DS worse, **Cervix**: 35.1 DS worse, **Polyp**: 36.9 DS worse). The poor performance of DSL was unexpected, given the excellent results in (Chang et al., 2023). We attribute this to untuned hyperparameters. We have checked that it learns to generate relatively realistic images (see Figure 8), on which the U-Net that it trains performs well, however it generalizes to real test images poorly.

We additionally did minimal manual hyperparameter tuning of DSL in the Polyp setting and were able to improve its performance by 9.2 DS on average ("DSL (lightly tuned)" in Table 8; we only changed the hyperparameters of the U-Net: we switched the optimizer to AdamW with default hyperparameters, switched step-wise learning rate schedule to cosine annealing, added random color jitter and random rotation augmentations).

While the performance of FedAvg and DSL could potentially be improved further via hyperparameter tuning, their default performance is subpar, highlighting the benefit of automatic methods such as HyFree-S3, the hyperparameter-free nature of which is its key benefit.

Table 6: Comparison to the federated learning baselines for the **Cervix** data. The results of HyFree-S3 for sites A/B correspond to syn-real A/B (see Section 4.1). Mean ± st. dev. are reported for 5 folds.

| Metric | Test site | Method | | | |
|--------|-----------|--------|--------|--------|--------|
| | | syn-real A | syn-real B | FedAvg | DSL |
| DS | A | $80.2 \pm 1.7$ | $78.5 \pm 2.1$ | $65.0 \pm 1.9$ | $42.8 \pm 10.4$ |
| | B | $74.1 \pm 1.8$ | $75.9 \pm 1.6$ | $67.7 \pm 2.3$ | $41.2 \pm 0.6$ |

Table 7: Comparison to the federated learning baselines for the **Lung** data. The results of HyFree-S3 for sites A/B correspond to syn-real A/B (see Section 4.1). Mean ± st. dev. are reported for 5 folds.

| Metric | Test site | Method | | | |
|--------|-----------|--------|--------|--------|--------|
| | | syn-real A | syn-real B | FedAvg | DSL |
| DS | A | $76.5 \pm 1.6$ | $76.3 \pm 1.7$ | $71.2 \pm 2.5$ | $43.8 \pm 13.4$ |
| | B | $77.5 \pm 1.2$ | $77.3 \pm 1.4$ | $71.9 \pm 1.2$ | $44.3 \pm 15.4$ |

Table 8: Comparison to the federated learning baselines for the **Polyp** data. The results of HyFree-S3 for sites A/B correspond to syn-real A/B (see Section 4.1). Mean ± st. dev. are reported for 5 folds.

| Metric | Test site | Method | | | | |
|--------|-----------|-----------|-----------|---------|---------|---------------------|
| | | syn-real A | syn-real B | FedAvg | DSL | DSL (lightly tuned) |
| DS | A | $90.3 \pm 1.4$ | $82.7 \pm 3.4$ | $54.3 \pm 3.8$ | $51.8 \pm 1.1$ | $60.7 \pm 2.7$ |
| | B | $81.1 \pm 7.4$ | $82.2 \pm 3.9$ | $55.2 \pm 10.1$ | $42.3 \pm 12.1$ | $51.9 \pm 7.4$ |

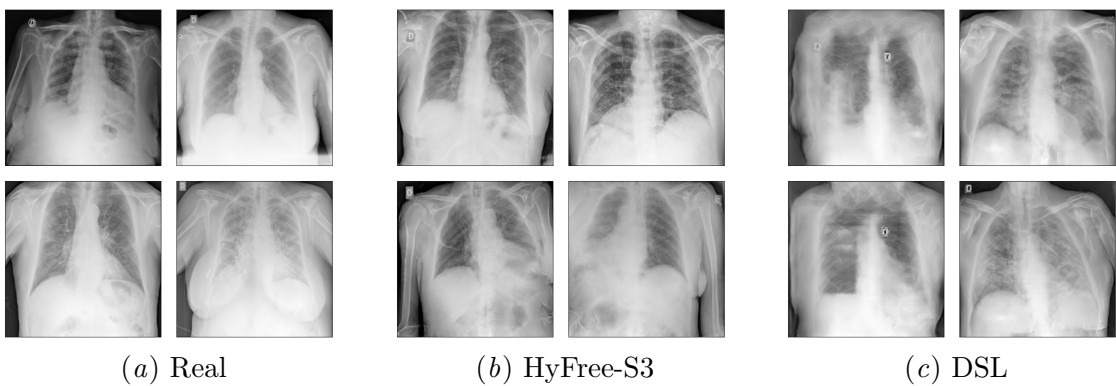

(*a*) Real           (*b*) HyFree-S3           (*c*) DSL

Figure 8: Random sample of **Lung** images (Site A, fold 0)

## Appendix E. Further discussion of memorization

We have been communicating with a data protection officer about the perspectives of synthetic data since this project started. Based on preliminary discussions, our method could greatly simplify data sharing between institutes and decrease privacy risks. However, currently, there are no official, internationally accepted, guidelines as to what constitutes memorization (of imaging data).

Therefore, it is difficult to say if our approach is "clinically acceptable", given lack of prior work on what that entails. While we are confident in our analysis, it could be made more robust by using several dissimilar embedding models, or by increasing the threshold. Still, our method relies on empirical performance of neural networks (as noted in Section 3.5), and therefore no mathematical guarantees can be given that no memorization occurs. Nonetheless, we believe that empirical evaluation similar to ours could be enough for clinical acceptance, once the guidelines are determined.

To further investigate the memorization phenomenon, we use an alternative method for finding memorized images from a concurrent work (Dar et al., 2024), where it was successfully used to find medical images memorized by a diffusion model. The method is similar to the approach we used in that it relies on embeddings of images via a neural net, with the two differences from our work being the model (the authors trained their own embedding model on the target dataset) and the similarity measure (the authors used correlation).

We apply this approach in the **Lung** setting. Whereas our method flagged close to zero samples as memorized, this approach flags $\approx 12\%$. However, visual inspection of the synthetic samples closest to some real ones (Figure 9) indicates that the synthetic samples are not duplicates of the real ones (based on our judgement). While similar, they differ in their details, and do not satisfy the properties on which the definition of memorization from Dar et al. (2024) is based: they are not variants of the original image derived via rotation, flipping, or contrast adjustment.

Nonetheless, memorization is difficult to define, and perhaps the eventual guidelines would enforce stricter definitions of memorization that would include these samples. If so, such synthetic outputs could be removed by using an appropriate embedding model within our method. HyFree-S3 is agnostic to the duplicates removal technique used within, better techniques can be substituted when they are developed.

## Appendix F. Pretraining using single-site synthetic data

Does pretraining on multi-site synthetic data have additional benefit over pretraining on single-site synthetic data? Here we compare our default setting (generating $10 \times n_{\text{real}}$ images at each site and pulling them together) to the setting of generating $20 \times n_{\text{real}}$ images at one site. In both settings, a U-Net is pretrained with the synthetic images and fine-tuned with the real local data. The experiments are performed for two sites with the **Cervix** data. Table 9 demonstrates that the networks pretrained with the local synthetic data (*syn-local-real*) achieve on average 2.7 DS worse performance than the networks pretrained on pooled synthetic data (*syn-real*). This result empirically demonstrates the benefit of bringing data from multiple sites together.

Table 9: Comparison of no pretraining (*real*) to pretraining on local synthetic data only (*syn-local-real*) or pooled synthetic data (*syn-real*) in the **Cervix** setting. Mean $\pm$ st. dev. are reported for 5 folds.

| Metric | Test site | Training setting | | | | | |
|---|---|---|---|---|---|---|---|
| | | real A | real B | syn-local-real A | syn-local-real B | syn-real A | syn-real B |
| DS | A | $80.1 \pm 1.9$ | $72.2 \pm 3.9$ | $80.0 \pm 1.8$ | $72.9 \pm 3.2$ | $80.2 \pm 1.7$ | $78.5 \pm 2.1$ |
| | B | $70.8 \pm 2.5$ | $73.7 \pm 4.4$ | $71.6 \pm 1.7$ | $73.3 \pm 3.8$ | $74.1 \pm 1.8$ | $75.9 \pm 1.6$ |

## Appendix G. Comparison to the standard StyleGAN2 architecture

To experimentally confirm that transferring the nnU-Net architectural parameters to Style-GAN2 is reasonable, we compare our GANs to the standard StyleGAN2 trained with square images of a resolution of a power of two. The experiment is performed with the Polyp-B data, for which the nnU-Net determined resolution ($388 \times 320$) is the farthest from a square where the dimensions of the sides are a power of two (note that for such images our Style-GAN2 architecture would be exactly the same as the standard one). The images are resized to the closest power of two ($256 \times 256$), and the standard StyleGAN2 is trained. Then synthetic images are generated and resized to $388 \times 320$. Afterwards, the Frechet Inception

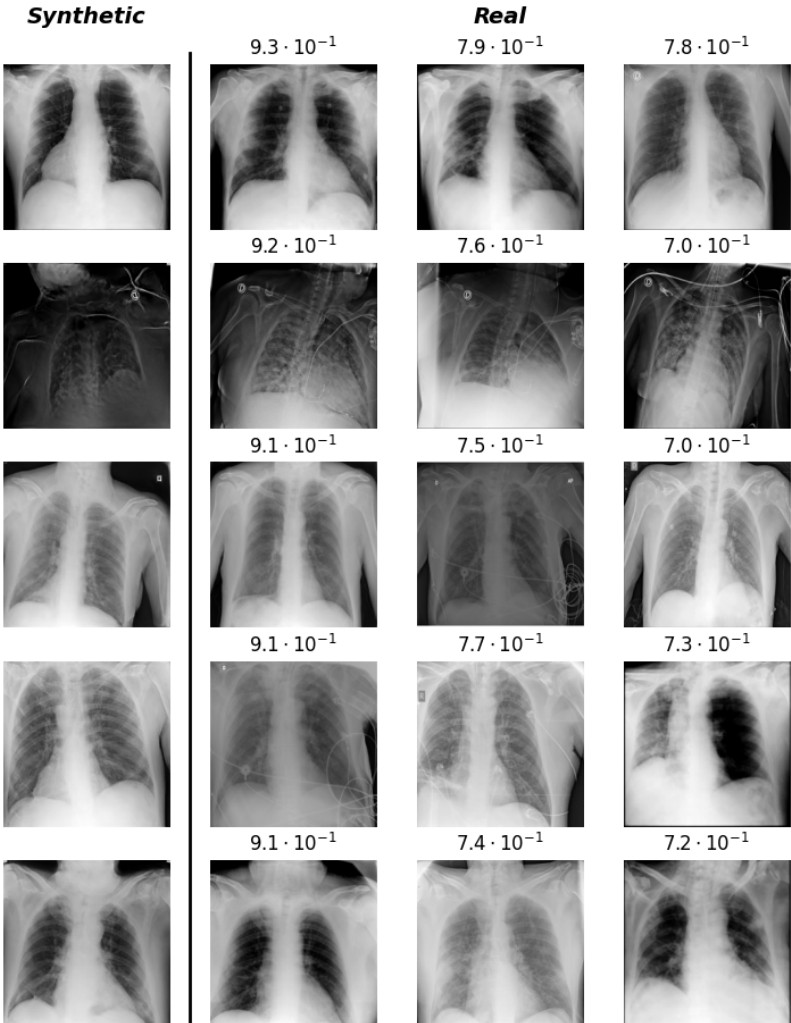

Figure 9: Alternative memorization detection: examples of synthetic images flagged as memorized and three nearest-neighbor real images for each (annotated with similarity to the synthetic image). The synthetic images with the highest similarity to real images are visualized.

Distance (FID) (Heusel et al., 2017) is calculated between them and the real images. FID is an established metric of GAN quality, lower values are better.

We compare the FID of the StyleGAN2-256 × 256 to our StyleGAN2-388 × 320, and find the latter to achieve better results ($102.4 \pm 4.0$ FID vs $88.0 \pm 3.2$ FID). This confirms that the architecture is reasonable, as it was able to make use of the increased resolution of its inputs to achieve higher generation quality.

