# OpenReview forum: "Hyperparameter-Free Medical Image Synthesis for Sharing Data and Improving Site-Specific Segmentation"
_MIDL.io/2024/Conference — MIDL 2024 Poster_

### Official Review · Reviewer_EtyB · 2024-02-27

**Confidence:** 4
**Preliminary Rating:** 1
**Recommendation:** Poster
**Final Rating:** 2

**Summary:**

This paper introduces an automatically adjustable StyleGAN2 setup and integrates it with nnU-Net to create a Hyperparameter-Free medical image Synthesis, Sharing, and Segmentation method called HyFree-S3. The method is evaluated on three segmentation settings.

**Strengths:**

The setting and the method seem interesting and the paper is easy to follow.
The method seems can protect the data privacy for collaborative learning.
The method is like a synthetic centralized learning compared to federated learning.

**Weaknesses:**

1.	Please discuss the federated learning in this paper, and compare their advantages and disadvantages. Because your setting is like a synthetic centralized learning to protect privacy, comparison with decentralized methods is also very important to position this setting.
2.	Motivation is still not clear. Why not just use one site to synthesize a large scale and diverse data pool, but use complicated multi-site collaboration to achieve the goals? So far, I am not convinced by the problem setting.
3.	Experiments should be improved: like question 2, use one site but generate more samples for training -> discard the collaborative training; federated learning should be compared.

**Detailed Comments:**

1.	Please discuss the federated learning in this paper, and compare their advantages and disadvantages. Because your setting is like a synthetic centralized learning to protect privacy, comparison with decentralized methods is also very important to position this setting.
2.	Motivation is still not clear. Why not just use one site to synthesize a large scale and diverse data pool, but use complicated multi-site collaboration to achieve the goals? So far, I am not convinced by the problem setting.
3.	Experiments should be improved: like question 2, use one site but generate more samples for training -> discard the collaborative training; federated learning should be compared.

**Justification Of Final Rating:**

The rebuttal solved most of my concerns, although I am still not so convinced by the problem setting in practice.
As for generating more samples for one site: I think the mentioned diversity limitation can be easily solved using advanced generative models. The domain shift of scanners can also be readily simulated, for example, by using some techniques in domain generalization like exchanging the style using FFT. Thus, It is hard to discern if the issue has been resolved because the added empirical results are possibly dependent on the limitation of the used generative methods. Or for GAN-based methods, you can use GAN to generate more samples and followed by a style transformation used in domain generalization for one site data.

**Justification Of The Preliminary Rating:**

Hard to discern the motivation of the problem setting and the method. Why not just use one site to synthesize a large scale and diverse data pool, but use complicated multi-site collaboration to achieve the goals?
Experiments should be improved.
Lack of federated learning discussion in this paper.

**Questions To Address In The Rebuttal:**

See above

**Special Issue:**

No

---

> ### Author Response · Authors · 2024-03-15
> **Official Comment by Authors (Part 1)**
>
> We thank the reviewer for their valuable feedback.
>
> > “1. Please discuss the federated learning in this paper, and compare their advantages and disadvantages. Because your setting is like a synthetic centralized learning to protect privacy, comparison with decentralized methods is also very important to position this setting.”
>
> We agree that federated learning is a relevant alternative to our approach, therefore the paper already includes a discussion of federated learning in the related work (section 2.2), as well as a comparison of the properties of our approach to prior work. The limitations of our method are discussed in the conclusion (section 5). If our discussion omits important details, we would appreciate them being pointed out so that we could include them.
>
> > “2. Motivation is still not clear. Why not just use one site to synthesize a large scale and diverse data pool, but use complicated multi-site collaboration to achieve the goals? So far, I am not convinced by the problem setting.”
>
> The motivation for using data from more than one site is that a single site has a limited amount of data, covering only a part of the true data distribution. While generative models could learn to generate an infinite amount of data similar to their training data, they will not generate data outside their training distribution. For example, a GAN trained on data collected with scanner X will never produce images similar to those collected with scanner Y (or: a GAN trained on the images of cats will never produce an image of a dog). At the same time, deep learning models benefit from an increased amount of diverse data. Therefore, training segmentation models on data from multiple sites should improve their performance and generality.
>
> > “3. Experiments should be improved: like question 2, use one site but generate more samples for training -> discard the collaborative training;”
>
> To compare generating more training samples from one site against generating samples from multiple sites, we run an additional experiment and report it in Appendix F. Specifically, we compare our default setting (generating 10 times the number of real images at each site and pulling them together) to the setting of generating 20 times the number of real images at one site. In both settings, a U-Net is pretrained with the synthetic images and fine-tuned with the real local data. The experiments are performed for two sites with the Cervix data (5 folds). The networks pretrained with the local synthetic data (syn-local-real) achieve on average 2.7 Dice Score (DS) worse performance than the networks pretrained on pooled synthetic data (syn-real). This serves as an empirical confirmation of our reasoning behind using multi-site synthetic data as outlined above.
>
> _(the response is continued in Part 2 due to character limit)_

---

> ### Author Response · Authors · 2024-03-15
> **Official Comment by Authors (Part 2)**
>
> _(this is the continuation of Part 1 of our response)_
>
> > “Federated learning should be compared.”
>
> To compare our approach to federated learning, we run two federated learning baselines: Federated Averaging (FedAvg) [1] (in combination with nnU-Net) and Distributed Synthetic Learning (DSL) [2], a state-of-the-art approach to training a GAN in a federated manner to be used for generating synthetic data and training a U-Net. Since the key contribution of our method is its automatic and hyperparameter-free nature, we run the baselines in a similar setting of no hyperparameter tuning (for FedAvg, we used 1000 communication rounds with 1 epoch of local training in between since it was the best setting in [1]; for DSL, we followed the paper and the official code base (https://github.com/tommy-qichang/DSL_All_Code), using the median values for hyperparameters that differed across the three datasets in DSL: λ-L1: 300, 150, 100 → 150; batch size: 6, 6, 3 → 6).
>
> The results of the experiments are reported in the Appendix D. FedAvg on average has 5.3 worse Dice Score (DS) (on the scale 0-100) than our HyFree-S3 in the i.i.d. setting of Lung, with the gap widening further in the non-i.i.d settings: 10.7 DS for Cervix, 29.2 DS for Polyp. DSL performs substantially worse in all settings (on average over 5 folds: Lung: 32.8 DS worse, Cervix: 35.1 DS worse, Polyp: 36.9 DS worse). The poor performance of DSL was unexpected, given the excellent results reported in [2]. We attribute this to untuned hyperparameters. We have checked that it learns to generate relatively realistic images (see Figure 8), on which the U-Net that it trains performs well, however it generalizes to real test images poorly. We additionally did minimal manual hyperparameter tuning of DSL in the Polyp setting and were able to improve its performance by 9.2 DS on average (see Appendix D for details).
>
> While the performance of FedAvg and DSL could potentially be improved further via hyperparameter tuning, their default performance is subpar, highlighting the benefit of automatic methods such as HyFree-S3, the hyperparameter-free nature of which is its key benefit.
>
> We updated our code to include the FedAvg implementation: https://github.com/AwesomeLemon/hyfree-s3-midl-review. For the DSL experiments, see our fork of the official code base: https://github.com/AwesomeLemon/DSL_All_Code.
>
> We believe that we addressed all the concerns of the reviewer. If any unclarities remain, we would be happy to clear them up.
>
> [1] McMahan, B., Moore, E., Ramage, D., Hampson, S., & Arcas, B. A. y. (2017). Communication-Efficient Learning of Deep Networks from Decentralized Data. Proceedings of the 20th International Conference on Artificial Intelligence and Statistics, 1273–1282. https://proceedings.mlr.press/v54/mcmahan17a.html
>
> [2] Chang, Q., Yan, Z., Zhou, M., Qu, H., He, X., Zhang, H., Baskaran, L., Al’Aref, S., Li, H., Zhang, S., & Metaxas, D. N. (2023). Mining Multi-Center Heterogeneous Medical Data with Distributed Synthetic Learning. Nature Communications, 14(1). https://doi.org/10.1038/s41467-023-40687-y

---

### Official Review · Reviewer_GiX9 · 2024-02-28

**Confidence:** 4
**Preliminary Rating:** 3
**Recommendation:** Poster
**Final Rating:** 4

**Summary:**

The authors propose a distributed learning approach that allows for data synthesis, even in data sensitive scenarios, with an application in image segmentations. They propose the training of an image synthesis model per site, which can be combined to generate synthetic data from all sites, containing site-specific but not sensitive data. The proposed method outperforms models trained on data from a single site.

**Strengths:**

HyFree-S3 is well-designed and intuitive, building on the well-founded basis of nnUNet. The authors evaluate on public datasets as well, helping reproducibility. The authors also propose an automated detection of memorization to prevent data leakage.

**Weaknesses:**

The authors show that their method significantly outperforms models trained on a single site. However, the real target would be to show that training on synthetic data is not significantly worse than training on all the real data available.

**Detailed Comments:**

- It is not stated, only implied: is it correct that real-all is always significantly better than all other models?

**Justification Of Final Rating:**

I would like to thank the authors for their thorough responses. I believe that my main concerns, and the concerns of the other reviewers have been adequately addressed, and taking all the review comments and responses into consideration, I am changing my final rating.

**Justification Of The Preliminary Rating:**

The proposed method in the manuscript has strong merit, the image synthesis models outperform certain baselines, however they do not reach the performance set out by the described problem. I believe further evaluations of the image synthesis models, the automated data leakage detection, and the fine-tuning stages would provide a stronger foundation of the study.

**Questions To Address In The Rebuttal:**

-  Using the pre-processing steps of nnUNet is certainly advantageous, however selecting the hyperparameters of the encoder and the decoder is more questionable. Although the feature space sizes of the encoder/decoder of a UNet and the generator/discriminator of a GAN are similar, their objectives are entirely different. The reviewer thinks it does not "appear natural to reuse the hyperparameters" determined by the nnUNet in the UNet architecture. Did the authors perform experiments to show that this is a reasonable decision?
- The results claim that "the images are broadly similar but clearly distinct and do not demonstrate memorization". I think it's important to note that this is according to the authors' definition of memorization based on OpenCLIP embeddings. What do the authors think about the clinical implications of this method? Do they believe that their results are "clinically acceptable"? Would it be allowed to use this threshold to distribute synthetic data generated by such a model? Or is there still a chance of memorization? I believe the approach is very interesting, however it should be evaluated further to be trustworthy.

---

> ### Author Response · Authors · 2024-03-15
> **Official Comment by Authors (Part 1)**
>
> We thank the reviewer for their valuable feedback.
>
> > “However, the real target would be to show that training on synthetic data is not significantly worse than training on all the real data available.”
>
> Achieving the performance of central training with all the real data without sharing the data would be ideal. However, the assumption of this research is that we _cannot_ bring real data together (as specified in section 3.1). Under this assumption, the practically valuable goal is to improve upon the baseline of using only local real data, which we achieve.
>
> > “It is not stated, only implied: is it correct that real-all is always significantly better than all other models?”
>
> This is generally correct, which we acknowledged in Section 4.3.1 (“While training on real data centrally (real-all) gives the best results overall, our method comes close”). The only exception is Polyp A, where syn-real-A is better by 0.3 Dice score, which is not a meaningful improvement. While we do not achieve the upper-bound results of the real-all performance, we believe that improving upon real-A and real-B is useful in practice, with further potential improvements left for future research.
>
> > “Using the pre-processing steps of nnUNet is certainly advantageous, however selecting the hyperparameters of the encoder and the decoder is more questionable. Although the feature space sizes of the encoder/decoder of a UNet and the generator/discriminator of a GAN are similar, their objectives are entirely different. The reviewer thinks it does not "appear natural to reuse the hyperparameters" determined by the nnUNet in the UNet architecture. Did the authors perform experiments to show that this is a reasonable decision?”
>
> We agree that the objectives of the encoder/decoder of a UNet and the discriminator/generator of a GAN are dissimilar when considered concretely, however we argue that they are, in fact, similar when considered at a higher level of abstraction. To improve clarity, we have rephrased our explanation in section 3.3, here we provide an extended version of it (and report experimental results afterwards):
>
> The goal of a GAN-Generator is to transform a low-resolution latent representation of an image into an image. Starting from a low-resolution tensor (shape C\*H\*W, where C is large, H and W are small), a GAN-Generator needs to gradually add information in each layer. It should simultaneously increase resolution (H, W) and decrease the number of channels (C). Similarly, the goal of a U-Net-Decoder is to transform a low-dimensional latent representation of an image into a mask (ignoring skip connections; see below). Similar to a GAN-Generator, it starts from a low-resolution tensor and needs to gradually increase the resolution and decrease the number of channels. The architectures of both GAN-Generator and U-Net-Decoder need to strike the correct balance between the speeds of increasing the resolution and decreasing the number of channels. As such, a network that strikes the correct balance in one setting, seems likely to do so in the other.
>
> Note that we only reuse the nnU-Net parameters related to this “speed of transformation” (kernel sizes, strides, network depths), and leave the rest of the StyleGAN2 architecture that is GAN-specific (e.g., the structure of synthesis layers) untouched. We acknowledge that the mapping between the GAN and the U-Net is imperfect (e.g., there are no skip connections in the GAN) but we hope we were able to better explain why the transfer of architectural parameters from one to the other appears natural to us.
>
> (Analogously, both the U-Net-Encoder and the GAN-Discriminator need to gradually decrease the spatial resolution and increase the number of channels to arrive at a low-dimensional representation)
>
> _(the response is continued in Part 2 due to character limit)_

---

> ### Author Response · Authors · 2024-03-15
> **Official Comment by Authors (Part 2)**
>
> _(this is the continuation of Part 1 of our response)_
>
> To **experimentally confirm** that transferring the nnU-Net architectural parameters to StyleGAN2 is reasonable, we compare our GANs to the standard StyleGAN2 trained with square images of a resolution of a power of two (see Appendix G). The experiment is performed with the Polyp-B data, for which the nnU-Net determined resolution (388\*320) is the farthest from a square where the dimensions of the sides are a power of two (note that for such images our StyleGAN2 architecture would be exactly the same as the standard one). The images are resized such that the sides are the closest power of two (256\*256), and the standard StyleGAN2 is trained. Then, synthetic images are generated and resized to 388\*320. Afterwards, the Frechet Inception Distance (FID) [1] is calculated between them and the real images. FID is an established metric of GAN quality, lower values are better.
>
> We compare the FID of the StyleGAN2-256x256 to our StyleGAN2-388x320, and find the latter to achieve better results (102.4 ± 4.0 FID vs 88.0 ± 3.2 FID). This confirms that the architecture is reasonable, as it was able to make use of the increased resolution of its inputs to achieve higher generation quality.
>
> Additional evidence that the GAN architectures are reasonable comes from our main experiments, where pretraining on the synthetic data produced by the GANs leads to improved performance on real data (this shows that the GAN architectures were successfully trained to capture the essence of the real data, which is their purpose).
>
> > The results claim that "the images are broadly similar but clearly distinct and do not demonstrate memorization". I think it's important to note that this is according to the authors' definition of memorization based on OpenCLIP embeddings.
>
> To clarify, in the context of that paragraph, we meant that no memorization is observed upon visual inspection, but given that the images were selected for visualization based on OpenCLIP embeddings, the reviewer is correct, we added the appropriate note to section 4.3.3.
>
> > What do the authors think about the clinical implications of this method? Do they believe that their results are "clinically acceptable"? Would it be allowed to use this threshold to distribute synthetic data generated by such a model? Or is there still a chance of memorization?
>
> We agree that these are all important questions, we have been communicating with a data protection officer about the perspectives of synthetic data since this project started. Based on preliminary discussions, our method could greatly simplify data sharing between institutes and decrease privacy risks. However, currently, there are no official, internationally accepted, guidelines as to what constitutes memorization (of imaging data).
>
> Therefore, it is difficult to say if our approach is "clinically acceptable", given lack of prior work on what that entails. We are confident in our analysis but it could be made more robust by using several dissimilar embedding models, or by increasing the threshold. Still, our method relies on empirical performance of neural networks, and therefore no mathematical guarantees can be given that no memorization occurs. Nonetheless, we believe that empirical evaluation similar to ours could be enough for clinical acceptance, once the guidelines are determined.
>
> As to the immediate implications, we believe our method to be beneficial, since the current standard practice seems to be to release synthetic data or generative models without any concern for memorization (e.g., [2, 3]).
>
> _(the response is continued in Part 3 due to character limit)_

---

> ### Author Response · Authors · 2024-03-15
> **Official Comment by Authors (Part 3)**
>
> _(this is the continuation of Part 2 of our response)_
>
>
>
> To further investigate the memorization phenomenon, we use an alternative method for finding memorized images from a concurrent work [4], where it was successfully used to find medical images memorized by a diffusion model. The method is similar to the approach we used in that it relies on embeddings of images via a neural net, with the two differences from our work being the model (the authors trained their own embedding model on the target dataset) and the similarity measure (the authors used correlation).
>
> We apply this approach in the Lung setting (see Appendix E). Whereas our method flagged close to zero samples as memorized, this approach flags ~12%. However, visual inspection of the synthetic samples closest to some real ones (Figure 9) indicates that the synthetic samples are not duplicates of the real ones (based on our judgement). While similar, they differ in their details, and do not satisfy the properties on which the definition of memorization from [4] is based: they are not variants of the original image derived via rotation, flipping, or contrast adjustment.
>
> Nonetheless, memorization is difficult to define, and perhaps the eventual guidelines would enforce stricter definitions of memorization that would include these samples. If so, such synthetic outputs could be removed by using an appropriate embedding model within our method. HyFree-S3 is agnostic to the duplicates removal technique used within, better techniques can be substituted when they are developed. As memorization is a complicated topic that is not the focus of our paper, we leave deeper investigation to future research.
>
> We believe that we addressed all the questions of the reviewer. If any remain, we would be glad to answer them.
>
> [1] Heusel, M., Ramsauer, H., Unterthiner, T., Nessler, B., & Hochreiter, S. (2017). GANs Trained by a Two Time-Scale Update Rule Converge to a Local Nash Equilibrium. Advances in Neural Information Processing Systems, 30. https://papers.nips.cc/paper_files/paper/2017/hash/8a1d694707eb0fefe65871369074926d-Abstract.html
>
> [2] Pinaya, W. H., Tudosiu, P. D., Dafflon, J., Da Costa, P. F., Fernandez, V., Nachev, P., ... & Cardoso, M. J. (2022, September). Brain imaging generation with latent diffusion models. In MICCAI Workshop on Deep Generative Models (pp. 117-126). Cham: Springer Nature Switzerland.
>
> [3] Hamamci, I. E., Er, S., Simsar, E., Tezcan, A., Simsek, A. G., Almas, F., ... & Menze, B. (2023). GenerateCT: Text-Guided 3D chest CT generation. arXiv preprint arXiv:2305.16037.
>
> [4] Dar, S. U. H., Seyfarth, M., Kahmann, J., Ayx, I., Papavassiliu, T., Schoenberg, S. O., & Engelhardt, S. (2024). Unconditional Latent Diffusion Models Memorize Patient Imaging Data (arXiv:2402.01054). arXiv. https://doi.org/10.48550/arXiv.2402.01054

---

> > ### Comment · Reviewer_GiX9 · 2024-03-21
> > **Response to the authors**
> >
> > I would like to thank the authors for their thorough response to my questions, and the questions of the other reviewers. I hope a conversation with Reviewer EtyB will happen during the rebuttal phase. I believe that their main concerns have all been adequately addresssed as well, therefore I am changing my rating appropriately.

---

> > > ### Author Response · Authors · 2024-03-22
> > > **Response to reviewer GiX9**
> > >
> > > We are glad that no further unclarities remain, and we thank the reviewer for helping us improve the paper.

---

### Official Review · Reviewer_rS4k · 2024-02-28

**Confidence:** 3
**Preliminary Rating:** 4
**Recommendation:** Poster
**Final Rating:** 4

**Summary:**

This paper presents a hyperparameter-free sitribution learning method for medical image synthesis, sharing, and segmentation. The method consists of a variant of StyelGAN2, motivated by nnU-Net, for hyperparameter-free data synthesis, a nnU-Net based segmentation model on merged synthetic data, and a strategy to recognize generated samples that may be a memorization of real samples. The experiment results show that the sharing of these synthetic data improves the segmentation performance on the local sites and approaches that can be obtained by training on combined real data.

**Strengths:**

The proposed method for distributed learning is simple yet demonstrates promising results empirically.  The customization to StyleGAN2 to allow hyperparamter free generation appears promising.

The  component that checks for memorization provides interesting insight and some assurance into the quality of the generated samples.

**Weaknesses:**

While the experimental results demonstrated the benefit of the proposed work to merge synthetic data for training, it’d be good to see comparisons to other image synthesis and federated learning works to provide empirical evidence for the limitations discussed in the related works section.

**Detailed Comments:**

A couple of image synthesis and federated learning works have been discussed in related works section. Although the authors emphasized the advantage of being hyperparamter-free and the concern of privacy leak as the motivation for the presented work, it would still be good to include some of these existing works to understand how their performance may be affected by these cited limitations.

**Justification Of Final Rating:**

The authors have addressed my previous main comments in terms of adding comparative analyses. I think the paper will have good discussion value for MIDL. I will thus stay with my original rating and recommend the acceptance of the paper.

**Justification Of The Preliminary Rating:**

This paper is overall well written and motivated, presenting a simple workflow that is empirically promising. The paper can be strengthened by additional comparative analyses to existing related works.

**Questions To Address In The Rebuttal:**

Adding some comparative analyses will largely strengthen the work.

**Special Issue:**

No

---

> ### Author Response · Authors · 2024-03-15
>
> We thank the reviewer for their valuable feedback.
>
> > “While the experimental results demonstrated the benefit of the proposed work to merge synthetic data for training, it’d be good to see comparisons to other image synthesis and federated learning works to provide empirical evidence for the limitations discussed in the related works section.”
>
> To compare our approach to federated learning, we run two federated learning baselines: Federated Averaging (FedAvg) [1] (in combination with nnU-Net) and Distributed Synthetic Learning (DSL) [2], a state-of-the-art approach to training a GAN in a federated manner to be used for generating synthetic data and training a U-Net. Since the key contribution of our method is its automatic and hyperparameter-free nature, we run the baselines in a similar setting of no hyperparameter tuning (for FedAvg, we used 1000 communication rounds with 1 epoch of local training in between since it was the best setting in [1]; for DSL, we followed the paper and the official code base (https://github.com/tommy-qichang/DSL_All_Code), using the median values for hyperparameters that differed across the three datasets in DSL: λ-L1: 300, 150, 100 → 150; batch size: 6, 6, 3 → 6).
>
> The results of the experiments are reported in the Appendix D. FedAvg on average has 5.3 worse Dice Score (DS) (on the scale 0-100) than our HyFree-S3 in the i.i.d. setting of Lung, with the gap widening further in the non-i.i.d settings: 10.7 DS for Cervix, 29.2 DS for Polyp. DSL performs substantially worse in all settings (on average over 5 folds: Lung: 32.8 DS worse, Cervix: 35.1 DS worse, Polyp: 36.9 DS worse). The poor performance of DSL was unexpected, given the excellent results reported in [2]. We attribute this to untuned hyperparameters. We have checked that it learns to generate relatively realistic images (see Figure 8), on which the U-Net that it trains performs well, however it generalizes to real test images poorly. We additionally did minimal manual hyperparameter tuning of DSL in the Polyp setting and were able to improve its performance by 9.2 DS on average (see Appendix D for details).
>
> While the performance of FedAvg and DSL could potentially be improved further via hyperparameter tuning, their default performance is subpar, highlighting the benefit of automatic methods such as HyFree-S3, the hyperparameter-free nature of which is its key benefit.
>
> We updated our code to include the FedAvg implementation: https://github.com/AwesomeLemon/hyfree-s3-midl-review. For the DSL experiments, see our fork of the official code base: https://github.com/AwesomeLemon/DSL_All_Code.
>
> We hope that we have addressed the weakness highlighted by the reviewer, and we would be glad to answer any other question the reviewer may have.
>
> [1] McMahan, B., Moore, E., Ramage, D., Hampson, S., & Arcas, B. A. y. (2017). Communication-Efficient Learning of Deep Networks from Decentralized Data. Proceedings of the 20th International Conference on Artificial Intelligence and Statistics, 1273–1282. https://proceedings.mlr.press/v54/mcmahan17a.html
>
> [2] Chang, Q., Yan, Z., Zhou, M., Qu, H., He, X., Zhang, H., Baskaran, L., Al’Aref, S., Li, H., Zhang, S., & Metaxas, D. N. (2023). Mining Multi-Center Heterogeneous Medical Data with Distributed Synthetic Learning. Nature Communications, 14(1). https://doi.org/10.1038/s41467-023-40687-y

---

### Author Response · Authors · 2024-03-15

We thank all the reviewers for their insightful comments. We have addressed each point raised by the reviewers in the individual responses below, here we provide a summary of the main changes in the updated version of the paper:
1. Added a comparison to two federated learning baselines (Federated Averaging, Distributed Synthetic Learning), see Appendix D.
2. Extended the discussion of memorization, see Appendix E.
3. Added a comparison of pretraining on single-site synthetic data to pretraining on multi-site synthetic data, see Appendix F.
4. Added a comparison with the standard StyleGAN2 that does not use the adaptive architecture parameters, see Appendix G.

---

### Meta-Review · Area_Chair_Hheb · 2024-04-04

**Recommendation:** Accept (Poster)
**Confidence:** 5

**Metareview:**

Most of the reviewers’ critiques were addressed in the authors’ feedback, including a comparison with other federated learning methods. Reviewers acknowledged that the paper has been revised sufficiently. Although a question still remained by a reviewer (using other image generation methods rather than the proposed method), the proposed idea seems to be still worth discussing in MIDL.

---

### Decision · Program_Chairs · 2024-04-06

Accept (Poster)